# Risk factors associated with mortality in hospitalized patients with SARS-CoV-2 infection. A prospective, longitudinal, unicenter study in Reus, Spain

**Simona Iftimie[1], Ana F. López-Azcona[1], Manuel Vicente-Miralles[1], Ramon Descarrega-Reina[1], Anna Hernández-Aguilera[2,3], Francesc Riu[3], Josep M. Simó[4], Pedro Garrido[5], Jorge Joven[2], Jordi Camps[id][2]\*, Antoni Castro[1]**

**1** Department of Internal Medicine, Hospital Universitari de Sant Joan, Institut d'Investigació Sanitària Pere Virgili, Universitat Rovira i Virgili, Reus, Spain, **2** Unitat de Recerca Biomèdica, Hospital Universitari de Sant Joan, Institut d'Investigació Sanitària Pere Virgili, Universitat Rovira i Virgili, Reus, Spain, **3** Department of Pathology, Hospital Universitari de Sant Joan, Institut d'Investigació Sanitària Pere Virgili, Universitat Rovira i Virgili, Reus, Spain, **4** Laboratori de Referència del Camp de Tarragona i Terres de l'Ebre, Reus, Spain, **5** Intensive Care Unit, Hospital Universitari de Sant Joan, Institut d'Investigació Sanitària Pere Virgili, Universitat Rovira i Virgili, Reus, Spain

\* jcamps@grupsagessa.com

## Abstract

Spain is one of the countries that has suffered the most from the impact of severe acute respiratory syndrome coronavirus 2 (SARS-CoV-2), the strain that causes coronavirus disease 2019 (COVID-19). However, there is a lack of information on the characteristics of this disease in the Spanish population. The objective of this study has been to characterize our patients from an epidemiological point of view and to identify the risk factors associated with mortality in our geographical area. We performed a prospective, longitudinal study on 188 hospitalized cases of SARS-Cov-2 infection in *Hospital Universitari de Sant Joan*, in Reus, Spain, admitted between 15th March 2020 and 30th April 2020. We recorded demographic data, signs and symptoms and comorbidities. We also calculated the Charlson and McCabe indices. A total of 43 deaths occurred during the study period. Deceased patients were older than the survivors (77.7 ± 13.1 *vs*. 62.8 ± 18.4 years; $p < 0.001$). Logistic regression analyses showed that fever, pneumonia, acute respiratory distress syndrome, diabetes mellitus and cancer were the variables that showed independent and statistically significant associations with mortality. The Charlson index was more efficient than the McCabe index in discriminating between deceased and survivors.

This is one of the first studies to describe the factors associated with mortality in patients infected with SARS-CoV-2 in Spain, and one of the few in the Mediterranean area. We identified the main factors independently associated with mortality in our population. Further studies are needed to complete and confirm our findings.

**Data Availability Statement:** All relevant data are within the manuscript and its Supporting Information files.

**Funding:** This study was supported by a grant from the Fundació la Marató de TV3 (201807-10), Barcelona, Spain to JC. The funder had no role in study design, data collection and analysis, decision to publish, or preparation of the manuscript.

**Competing interests:** The authors have declared that no competing interests exist.

## Introduction

In January 2020, a new type of coronavirus was identified as the causative factor in a series of cases of severe pneumonia in the city of Wuhan, province of Hubei, in the People's Republic of China [1]. The World Health Organization gave the official name 'COVID-19' for this coronavirus disease, as well as the term 'severe acute respiratory syndrome coronavirus 2' (SARS-CoV-2) for the virus [2]. This virus is currently the cause of a global pandemic, producing hundreds of thousands of hospital admissions and deaths, with enormous effects on the health and life of the population and serious economic consequences for society. On 1$^{st}$ February, 2020, the first case of a SARS-CoV-2 positive patient in Spain was reported on the island of La Gomera [3] and, following that, the first cases diagnosed in the autonomous region of Catalonia date from 5$^{th}$ March [4]. The incubation period for SARS-CoV-2 ranges from 5 to 6 days on average, with cases being possible from 0 to 14 days [5]. The most common period of transmission of the virus begins 1–2 days before the onset of symptoms, and lasts for up to 5–6 days after the onset of symptoms [6]. The basic reproductive rate $R$ (the average of new cases secondary to a primary case) in our country is, at the time of writing, estimated to be <1; globally, the $R$ number ranges from 0 to 6 depending on various factors, in particular the political and public health measures imposed by the various governments that focus on complete cleaning of public spaces and a decrease in contact between individuals [7]. Identifying the epidemiological characteristics of this disease will help appropriate decisions to be made and thus to control the epidemic. Certain clinical symptoms of COVID-19 have been reported previously. The most frequent are: fever, dry cough, asthenia, expectoration, dyspnea, sore throat, headache, myalgia, arthralgia, chills, nausea or vomiting, nasal congestion, diarrhea, hemoptysis and conjunctival congestion (from highest to lowest frequency) [8,9]. Occasionally, symptoms of a different nature appear: neurological, such as altered consciousness or dizziness; cardiological, such as acute myocardial damage or heart failure; or ophthalmological, such as dry eye, blurred vision, foreign body sensation and conjunctival congestion [10–13].

To date, there is still a lack of information on the characteristics of SARS-CoV-2 infection outside China. Spain is one of the Western European countries that has suffered the most from the impact of COVID-19 and this pandemic has had a great impact on our public health system. The present study reports the results of an analysis of all cases hospitalized in the *Hospital Universitari de Sant Joan*, which is affiliated to the *Universitat Rovira i Virgili*, in Reus, Catalonia, Spain. The objective of the present study has been to characterize our patients' epidemiology and to identify the risk factors associated with mortality for this disease in our geographical area.

## Materials and methods

### Study design

This is a prospective longitudinal study conducted on all hospitalized cases of SARS-CoV-2 infection in *Hospital Universitari de Sant Joan*, in Reus, Spain admitted between 15$^{th}$ March 2020 and 30$^{th}$ April 2020. This hospital has 392 beds provided for hospitalization and social health care and is part of the Hospital Network for Public Use in Catalonia. It acts as a general hospital for a population of over 175,000 inhabitants, including primary care centers and residences for the elderly in the area. It is a reference center for the specialities of Oncology and Radiotherapy for the whole of the Tarragona province, which has a population of 550,000 inhabitants. SARS-CoV-2 infection was confirmed by reverse transcription-polymerase chain reaction (RT-PCR) using swab samples from the upper respiratory tract (nasopharyngeal/oropharyngeal exudate), from the lower respiratory tract (sputum/endotracheal aspirate/

bronchoalveolar lavage/bronchial aspirate) or from the lower digestive tract (rectal smear). Tests were carried out with the VIASURE *SARS-CoV-2* Real Time PCR Detection Kit that detects *ORF1ab* and *N* genes (CerTest Biotec, Zaragoza, Spain). RNA was extracted in a QIA-cube apparatus with RNeasy reagents (Qiagen N.V., Hilden, Germany) according to the manufacturer's instructions, and analyses were carried out in a 7500 Fast RT-PCR System (Applied Biosystems, Foster City, CA,USA). We recorded demographic data, comorbidities, and other acute or chronic infections. We also calculated the McCabe score as an index of clinical prognosis [14] and the Charlson index (abbreviated version) as a way of categorizing a patient's comorbidity [15]. The only inclusion criterion was to be a hospitalized patient with an analytical diagnosis of SARS-CoV-2. We excluded hospitalized patients with suspected SARS-CoV-2 infection but without laboratory confirmation, or patients who did not require hospitalization, with or without laboratory diagnosis of SARS-CoV-2 infection. Thirty-four patients required transfer to the Intensive Care Unit based on the attending specialist's criteria, and taking into account the CURB65 scale and the ATS/IDSA criteria [16,17]. This study was approved by the *Comitè d'Ètica i Investigació en Medicaments* (Institutional Review Board) of *Hospital Universitari de Sant Joan* (Resolution CEIM 040/2018, amended on 16 April 2020).

### Statistical analyses

Data are shown as means and standard deviations or as numbers and percentages. Statistical comparisons between two groups were carried out with the Student's *t* test (quantitative variables) or the $\chi$-square test (categorical variables). Logistic regression models were fitted to investigate the combined effect of selected variables on mortality. The diagnostic accuracy of the McCabe and Charlson indices in predicting mortality was assessed by receiver operating characteristics (ROC) analysis [18]. Statistical significance was set at $p \leq 0.05$. All calculations were made using the SPSS 25.0 statistical package (SPSS Inc., Chicago, IL, USA).

## Results

The raw data for this article are shown as Supporting Information. During the study period, a total of 188 patients were hospitalized for SARS-CoV-2 infection. The mean age was 66.4 ± 18.4 years (Range: 0–102) and a small majority were men (55.8 *vs.* 44.2%; $p < 0.001$). One hundred and eighteen patients were admitted to the Department of Internal Medicine, 34 to the Intensive Care Unit, and 36 to the Social Health Unit. Thirty-two patients were admitted to hospital due to causes unrelated to the suspicion of COVID-19 infection but gave a positive result in the RT-PCR. A total of 43 deaths occurred during the entire study period (Fig 1), so the case fatality rate was 22.9% based on the total number of COVID-19 hospitalized patients.

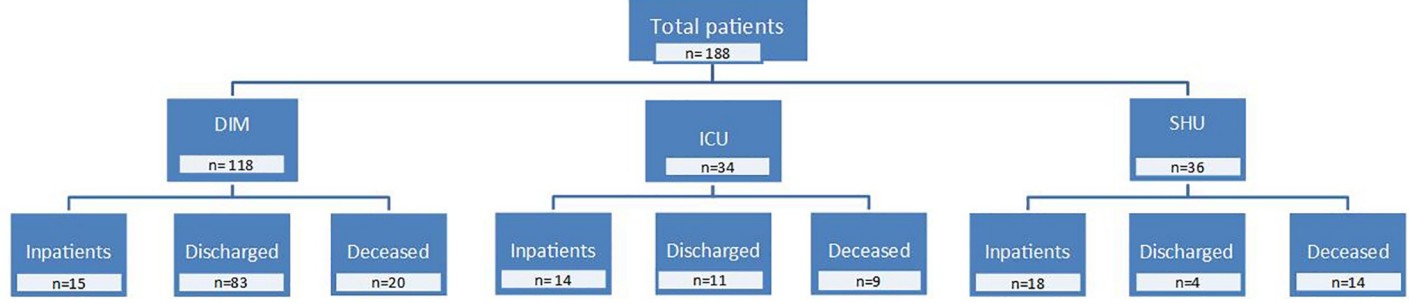

**Fig 1. Flow chart showing the distribution of hospitalized patients and the evolution of their disease.** DIM, Department of Internal Medicine; ICU, Intensive Care Unit; SHU, Social Health Unit.

Deceased patients were significantly older than the survivor patients (77.7 ± 13.1 *vs*. 62.8 ± 18.4 years; $p < 0.001$). Thirty-seven patients died of respiratory failure, 4 of multi-organ failure and 2 of cardiogenic shock. A total of 125 patients (66.5%) had chronic underlying diseases. Some seriously ill patients could not be admitted to the ICU due to their pathological history and/or comorbidities associated with their advanced age and who made aggressive treatments inadvisable.

The relationships between COVID-19 and the demographic and clinical variables are shown in Table 1 and Fig 2. Most of the cases and deaths were of patients between 70 and 89 years old (Fig 2A). The signs and symptoms present in more than 50% of the patients were, in descending order, fever (64.9%), dyspnea (58.0%), pneumonia (57.4%), and cough (51.6%) (Fig 2B). The most relevant comorbidities were cardiovascular diseases (50.5%), type 2 diabetes mellitus (26.0%), and chronic neurological diseases (19.1%) (Fig 2C). We also evaluated whether patients had had any behaviour that might be considered risky in the days prior to admission, and we observed that a high proportion of patients had attended another health center in the previous month or had been in contact with people infected with SARS-CoV-2 or with respiratory problems over the previous 14 days (Fig 2F). Five employees of our institution or the associated residences were hospitalized for COVID-19, although not requiring either intensive measures or ventilatory support.

Most of the patients presented low values on the Charlson and McCabe indices and, as expected, higher scores were associated with higher mortality (Fig 2D and 2E). When comparing the diagnostic accuracy of the ROC curves of these indices in their ability to discriminate between deceased patients and survivors, we found that Charlson index was more efficient, with higher values of the area under the curve (Fig 3).

Finally, since the different symptoms and comorbidities can be mutually interdependent and present cause-effect relationships between them, we wanted to identify which factors were independently associated with mortality. Logistic regression analyses showed that the presence of fever, pneumonia, acute respiratory distress syndrome, type 2 diabetes mellitus and cancer were the only variables that showed an independent and statistically significant association with mortality when they were adjusted for differences in age, gender, smoking status and alcohol intake (Tables 2 and 3).

One hundred and thirty-seven patients (72.9%) required one or more than one type of respiratory intervention, including noninvasive (face mask) or invasive (endotracheal tube) mechanical ventilation, high flow oxygen therapy (up to 60 L/min.) or conventional oxygen therapy (Table 4). No significant differences in mortality were observed in patients requiring globally analyzed respiratory intervention (25.5 *vs*. 17.6%, $p = 0.173$), but there was a non-significant trend towards higher mortality in the subgroup of patients receiving high flow oxygen therapy (38.9 *vs*. 21.8%, $p = 0.094$). Moreover, we did not find any significant difference in mortality in relation to whether patients were treated with anticoagulants or corticosteroids or not (Anticoagulants: 24.1 *vs*. 17.6%, $p = 0.399$; Corticosteroids: 26.0 *vs*. 21.8%, $p = 0.318$).

## Discussion

We carried out a RT-PCR determination for all the patients admitted to our center, regardless of the diagnosis. Thirty-two patients (17.0%) admitted for reasons other than suspected SARS-CoV-2 infection gave a positive result despite not presenting any symptoms. We believe that this is important since it highlights the need to perform diagnostic tests for this disease in all hospitalized patients, something which has not been given sufficient attention in the scientific literature.

Most of our patients were over 60 years old and mortality was very high (47.0%) among those over 80 years old. These results are consistent with those published so far, which show

**Table 1. Demographic and clinical characteristics of patients with COVID-19 infection.**

| Feature | Cases, n (%) |
| --- | --- |
| **Age, years** | |
| 0–9 | 1 (0.53) |
| 10–19 | 3 (1.60) |
| 20–29 | 4 (2.13) |
| 30–39 | 9 (4.79) |
| 40–49 | 13 (6.91) |
| 50–59 | 29 (15.43) |
| 60–69 | 39 (20.74) |
| 70–79 | 39 (20.74) |
| 80–89 | 41 (21.81) |
| 90–99 | 9 (4.79) |
| 100–109 | 1 (0.53) |
| **Gender** | |
| Male | 105 (55.8) |
| Female | 83 (44.2) |
| **Smoking status** | |
| No | 145 (77.1) |
| Yes | 9 (4.8) |
| Ex-smoker | 34 (18.1) |
| **Alcohol consumption** | |
| No | 179 (95.2) |
| Yes | 9 (4.8) |
| **Signs and symptoms** | |
| Fever | 122 (64.9) |
| Dyspnea | 109 (58.0) |
| Pneumonia | 108 (57.4) |
| Cough | 97 (51.6) |
| Chills | 42 (22.3) |
| Diarrhea | 42 (22.3) |
| Acute kidney failure | 18 (9.6) |
| Odynophagia | 13 (6.9) |
| Acute respiratory distress syndrome | 10 (5.3) |
| Vomiting | 9 (4.8) |
| Other respiratory symptoms | 7 (3.7) |
| **Disease risk factors** | |
| Cardiovascular disease (including hypertension) | 95 (50.5) |
| Type 2 diabetes mellitus | 49 (26.0) |
| Chronic neurological disease | 36 (19.1) |
| Chronic lung disease | 27 (14.4) |
| Chronic kidney disease | 27 (14.4) |
| Cancer | 26 (13.8) |
| Postpartum (< 6 weeks) | 2 (1.0) |
| Chronic liver disease | 2 (1.1) |
| Pregnancy | 1 (0.5) |
| **Risky contacts** | |
| Visit to another medical center last month | 73(38.8) |
| Contact with SARS-CoV-2 positive last 14 days | 55 (29.3) |

(*Continued*)

**Table 1.** (Continued)

| Feature | Cases, n (%) |
|---|---|
| Contact with respiratory infection last 14 days | 54 (28.7) |
| Travel in the last month | 25 (13.3) |
| Health worker | 5 (2.7) |
| **Charlson index** | |
| 0 | 81 (43.1) |
| 1 | 40 (21.3) |
| 2 | 42 (22.3) |
| 3 | 16 (8.5) |
| 4 | 8 (4.2) |
| 5 | 1 (0.5) |
| **McCabe index** | |
| Nonfatal disease | 133 (70.7) |
| Ultimately fatal disease | 45 (23.9) |
| Rapidly fatal disease | 10 (5.3) |
| **Mean days of admission** | 14 |
| **Discharges** | 98 |
| **Deaths** | 43 |

that age is one of the most important risk factors for COVID-19 [19–22]. It is accepted that age is a risk factor for respiratory diseases [19,23,24] and impairment of immune function associated with age has been identified as a major cause of high mortality due to severe pneumonia [23]. We observed a higher mortality rate in patients treated with high flow oxygen therapy, but the observed differences did not reach statistical significance. The limited size of our sample prevents us from obtaining reliable conclusions in this regard. Among the signs and symptoms of the disease, we found that fever, pneumonia, and acute respiratory distress syndrome were the only factors independently associated with mortality when adjusted for age, smoking and alcohol intake. These factors are among those that have been most frequently found in patients with COVID-19 in most of the studies conducted in China [19,25,26]. We did not observe any independent relationship between cough, chills or gastrointestinal disturbances and mortality, despite being present in a relatively high proportion of subjects, something which differs from what has been published previously [19].

The comorbidities showing a significant relationship with mortality were type 2 diabetes mellitus and cancer. We did not find any independent association with any other chronic metabolic disease, such as cardiovascular disease or others. The univariate analysis showed a high number of patients with these chronic alterations and the logistic regression analysis identified diabetes as the most relevant. Indeed, all of these metabolic diseases are closely related. Diabetes is a causative factor of hypertension and metabolic syndrome and these, in turn, can cause heart, vascular, liver, neurological and kidney diseases. Our study therefore suggests that diabetes might be a triggering factor for these disorders and therefore is related to mortality in patients infected with SARS-CoV-2. Type 2 diabetes mellitus has also been reported to be one of the most important factors related with COVID-19 severity in previous investigations conducted in China, Israel and Italy [25,27–29]. Indeed, the Italian study reported that 2/3 of the patients who died were diabetic [29]. Furthermore, diabetes is linked to a higher mortality in other viral infections, such as those caused by influenza A(H1N1), MERS-CoV and SARS-CoV viruses [30,31]. We also found a close relationship between cancer and COVID-19 mortality. One aspect that caught our attention is that, despite our hospital being the reference center for

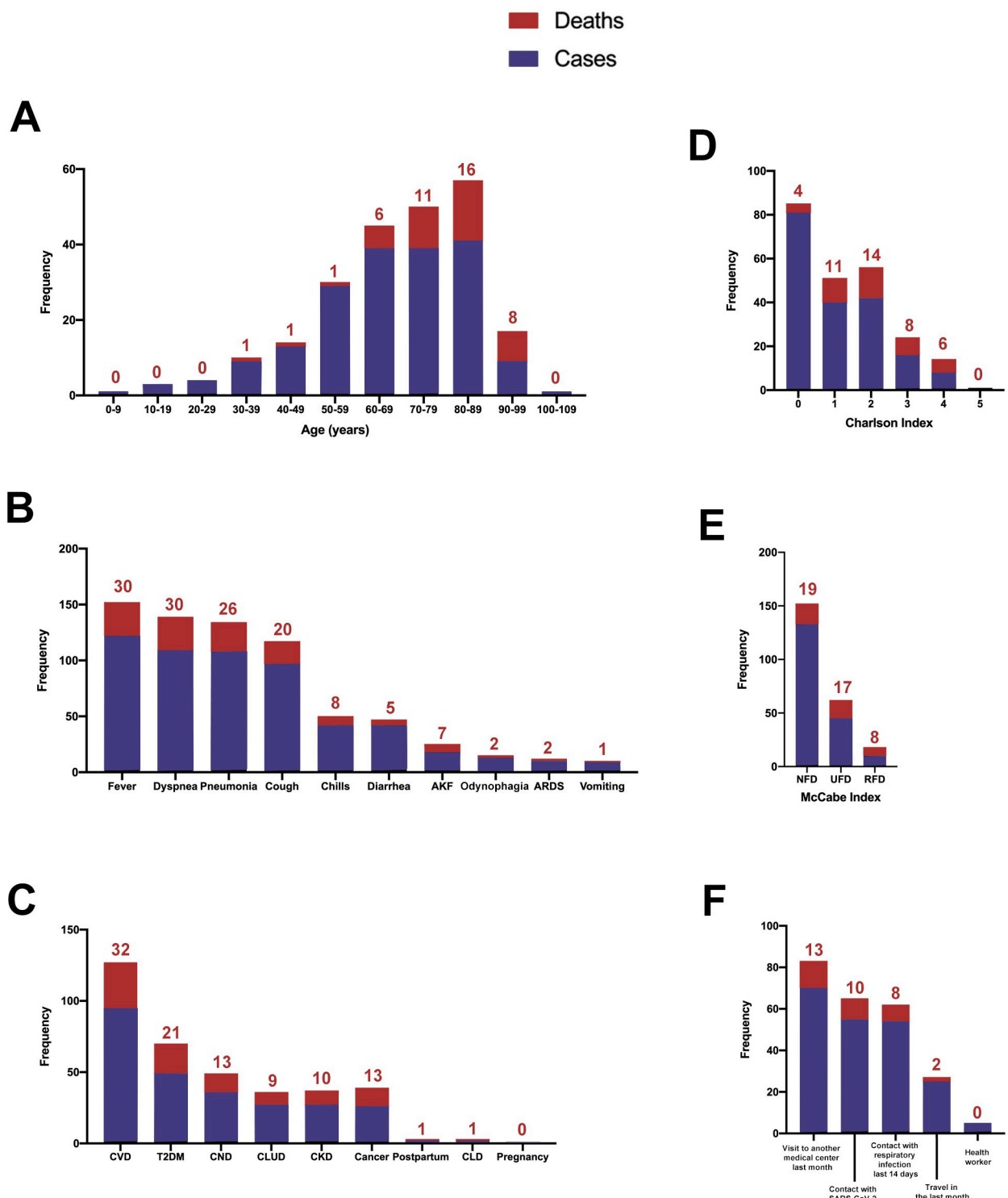

**Fig 2. Distribution of ages, clinical variables, and risk factors among patients with SARS-CoV-2 infection.** The numbers above the bars indicate the number of deceased patients. AKF, acute kidney failure; ARDS, acute respiratory distress syndrome; CKD, chronic kidney disease; CLD, chronic liver disease; CLUD, chronic lung disease; CND, chronic neurological disease; CVD, cardiovascular disease; NFD, non-fatal disease; RFD, rapidly fatal disease; T2DM, type 2 diabetes mellitus; UFD, ultimately fatal disease.

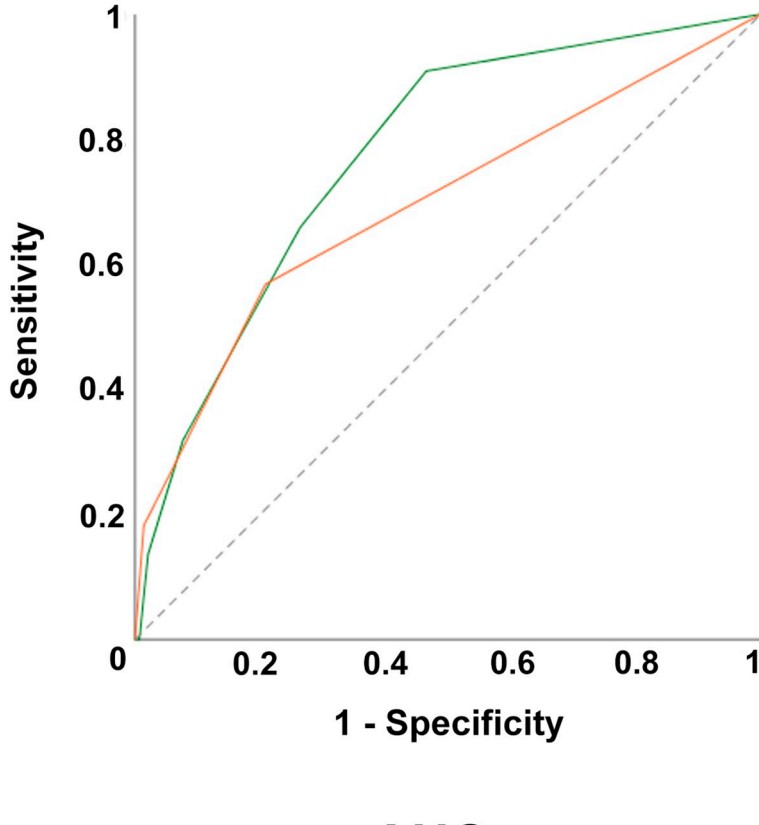

**AUC**

— **Charlson Index: 0.774 (0.698 - 0.849)**

— **McCabe Index: 0.695 (0.598 - 0.792)**

**Fig 3. Receiver operating characteristics (ROC) plots of Charlson and McCabe indices in COVID-19 patients and segregated with respect to mortality.** AUC, area under the curve.

Oncology in our province, the number of cancer patients infected with COVID-19 was relatively low. It might be that the corticosteroids often prescribed for the treatment of these patients offered some protection, as some studies have suggested [32,33]. However, we did not observe any significant difference in mortality in relation to whether patients were treated with corticosteroids or not. An alternative explanation is that, perhaps, these patients were more careful than the general population during confinement, which unfortunately cannot be proven. Having said that, the relationship between cancer and the mortality of our patients was evident. Patients with cancer are often immunosuppressed and as a result they are more likely to worsen rapidly if infected by SARS-Cov-2. Wuhan studies report that the incidence of cancer is higher in COVID-19 patients than in the general population [34,35]. However, definitive conclusions on this issue are hampered by the small sample size, the retrospective nature of most studies, the limited follow-up duration, and the heterogeneity of the disease and treatment strategies [36,37].

The influence of smoking on COVID-19 is controversial. An unusually low prevalence of current smoking among infected patients was observed in China [38] and the plausibility of using medicinal nicotine to lower infection and mitigate disease severity has been proposed

**Table 2. Logistic regression analysis on the relationships of signs and symptoms with deaths for COVID-19[a].**

| Variable | B | SE | Exp (B) | *p*-value |
|---|---|---|---|---|
| Fever | 1.107 | 0.554 | 3.024 | 0.046 |
| Cough | 0.068 | 0.544 | 1.070 | 0.901 |
| Pneumonia | -1.167 | 0.579 | 0.311 | 0.044 |
| Odynophagia | -1.473 | 1.044 | 0.229 | 0.159 |
| Chills | -0.897 | 0.675 | 0.408 | 0.184 |
| Acute respiratory distress syndrome | 3.074 | 1.010 | 21.636 | 0.002 |
| Other respiratory symptoms | 1.084 | 0.566 | 2.956 | 0.083 |
| Vomiting | -0.617 | 1.265 | 0.539 | 0.625 |
| Diarrhea | -0.712 | 0.595 | 0.491 | 0.232 |
| Age | 0.085 | 0.019 | 1.088 | <0.001 |
| Gender | 0.884 | 0.511 | 2.420 | 0.084 |
| Smoking status | -0.393 | 0.545 | 0.675 | 0.471 |
| Alcohol status | 0.571 | 0.807 | 1.769 | 0.479 |
| Constant | -8.323 | 1.644 | 0.000 | < 0.001 |

[a]Model summary: log-likelihood(-2) = 145.848; $r^2$ Cox & Snell = 0.268; $r^2$ Nagelkerke = 0.405; $p$ <0.001. B: Non-standardized β coefficient. SE: Standard error of B.

**Table 3. Logistic regression analysis on the relationships of comorbidities with deaths for COVID-19[a].**

| Variable | B | SE | Exp (B) | *p*-value |
|---|---|---|---|---|
| Type 2 diabetes mellitus | 0.914 | 0.424 | 2.493 | 0.031 |
| Cardiovascular diseases | 0.175 | 0.476 | 1.191 | 0.714 |
| Chronic liver diseases | -0.958 | 1.287 | 0.384 | 0.457 |
| Chronic lung diseases | 0.249 | 0.562 | 1.282 | 0.658 |
| Chronic kidney diseases | -0.301 | 0.539 | 0.740 | 0.576 |
| Chronic neurological diseases | 0.109 | 0.483 | 1.115 | 0.822 |
| Cancer | 1.313 | 0.506 | 3.719 | 0.009 |
| Age | 0.064 | 0.019 | 1.066 | 0.001 |
| Gender | 1.077 | 0.465 | 2.936 | 0.021 |
| Smoking status | -0.474 | 0.551 | 0.622 | 0.390 |
| Alcohol status | -0.148 | 0.801 | 0.862 | 0.853 |
| Constant | -7.010 | 1.441 | 0.001 | < 0.001 |

[a]Model summary: log-likelihood(-2) = 158.620; $r^2$ Cox & Snell = 0.217; $r^2$ Nagelkerke = 0.327; $p$ <0.001. B: Non-standardized β coefficient. SE: Standard error of B.

**Table 4. Selected treatments in patients with COVID-19 infection.**

| Treatment | Cases, n (%) |
|---|---|
| **Respiratory intervention** | 137 (72.9) |
| Noninvasive mechanical ventilation | 7 (3.7) |
| Invasive mechanical ventilation | 27 (14.4) |
| High flow oxygen therapy | 18 (9.6) |
| Conventional oxygen therapy | 137 (72.9) |
| **Anticoagulants[a]** | 170 (90.4) |
| **Corticosteroids[b]** | 73 (39.0) |

[a] Low-molecular-weight heparin.

[b] Methylprednisolone, dexamethasone, hydrocortisone or prednisone.

[39]. However, other studies indicate that smokers might be at higher risk because nicotine can directly impact the putative receptor for the virus (angiotensin-converting enzyme 2) and lead to harmful signaling in lung epithelial cells [40]. In the present study, we have found no firm positive or negative relationship between tobacco use and mortality because only 9 patients (4.8%) were active smokers at the time of the study. That might be explained by their generally advanced age and because many of them were suffering from chronic ailments that had advised them to quit tobacco use.

A novel aspect of our study has been to investigate the usefulness of some frequently used clinical scores in the evaluation of infectious diseases. We found that the Charlson index, which categorizes comorbidity might be more useful than the McCabe index in predicting death in these patients. We chose these indices because they are easy to apply in all patients, both in the less severe, and in those receiving palliative treatment or admitted to the Social Health Unit. Other scores are difficult to apply in less severe patients. For example, the CURB 1 scale only applies to patients with pneumonia [41]. The quick SOFA index requires measurement of respiratory rate, a data not frequently collected in milder patients [42], and SOFA and SAPS-II require arterial blood gas analysis, which is difficult to justify in patients not admitted to the ICU [43]. A limitation of the present study is the small sample size. Ours is not a big hospital and covers a relatively small geographical area. However, we believe that the results obtained are relevant since they might be representative of many similar centers in Western Europe and in the Mediterranean area, and little information is yet available on this issue.

## Conclusion

This is one of the first studies to describe the factors related with death in patients infected with SARS-CoV-2 in Spain, and one of the few from the Mediterranean basin. Our results identify age, fever, pneumonia, acute respiratory distress syndrome, type 2 diabetes mellitus and cancer as independent factors predicting lethality. Further studies are needed in similar centers to complete and confirm our findings.

## Supporting information

**S1 File.**
(SAV)

## Acknowledgments

The authors are indebted to all the staff of the *Hospital Universitari de Sant Joan*, doctors, nurses, assistants, cleaning and security personnel, and all the volunteer students, who with their enormous effort are managing to overcome this dramatic situation. Editorial assistance was provided by Phil Hoddy at the Service of Linguistic Resources of the *Universitat Rovira i Virgili*.

## Author Contributions

**Conceptualization:** Simona Iftimie, Jordi Camps.

**Data curation:** Simona Iftimie, Ana F. López-Azcona, Manuel Vicente-Miralles, Ramon Descarrega-Reina, Anna Hernández-Aguilera, Jordi Camps.

**Formal analysis:** Simona Iftimie, Anna Hernández-Aguilera, Jordi Camps.

**Funding acquisition:** Simona Iftimie, Jordi Camps, Antoni Castro.

**Investigation:** Simona Iftimie, Ana F. López-Azcona, Manuel Vicente-Miralles, Ramon Descarrega-Reina, Anna Hernández-Aguilera, Francesc Riu, Josep M. Simó, Pedro Garrido, Jorge Joven, Jordi Camps, Antoni Castro.

**Methodology:** Simona Iftimie, Ana F. López-Azcona, Manuel Vicente-Miralles, Ramon Descarrega-Reina, Anna Hernández-Aguilera, Jordi Camps.

**Project administration:** Simona Iftimie, Jordi Camps.

**Resources:** Simona Iftimie, Francesc Riu, Josep M. Simó, Pedro Garrido, Jorge Joven, Jordi Camps, Antoni Castro.

**Software:** Anna Hernández-Aguilera.

**Supervision:** Simona Iftimie, Jordi Camps.

**Validation:** Simona Iftimie, Jordi Camps.

**Writing – original draft:** Simona Iftimie, Manuel Vicente-Miralles, Anna Hernández-Aguilera, Jordi Camps.

**Writing – review & editing:** Simona Iftimie, Ana F. López-Azcona, Manuel Vicente-Miralles, Anna Hernández-Aguilera, Jordi Camps.

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
