## [Decision Letter · Decision Letter 0]

29 Jun 2020

PONE-D-20-15853

Risk factors associated with mortality in hospitalized patients with SARS-CoV-2 infection. A prospective, longitudinal, unicenter study in Reus, Spain

PLOS ONE

Dear Dr. Camps,

Thank you for submitting your manuscript to PLOS ONE. After careful consideration, we feel that it has merit but does not fully meet PLOS ONE’s publication criteria as it currently stands. Therefore, we invite you to submit a revised version of the manuscript that addresses the points raised during the review process.

 I have received the comments of the reviewers on your manuscript. The specific comments of the reviewers are included below. Please provide point by point response in your revised manuscript.

We look forward to receiving your revised manuscript.

Kind regards,

Muhammad Adrish

Academic Editor

PLOS ONE

Journal Requirements:

2. Thank you for including your ethics statement: "This study was approved 100 by the Ethics Committee of our

101 Institution (Resolution CEIM 040/2018, amended on 16 April 2020)."

"This study was supported by a grant from the Fundació la Marató de TV3 (201807-10), Barcelona, Spain."

4. Please upload a copy of Supporting Information which you refer to in your text on page 12.

Reviewers' comments:

Reviewer's Responses to Questions

**Comments to the Author**

1. Is the manuscript technically sound, and do the data support the conclusions?

Reviewer #1: Partly

Reviewer #2: Yes

2. Has the statistical analysis been performed appropriately and rigorously? 

Reviewer #1: Yes

Reviewer #2: Yes

3. Have the authors made all data underlying the findings in their manuscript fully available?

Reviewer #1: Yes

Reviewer #2: Yes

4. Is the manuscript presented in an intelligible fashion and written in standard English?

Reviewer #1: Yes

Reviewer #2: Yes

5. Review Comments to the Author

Reviewer #1: Authors present a prospective longitudinal study on 188 hospitalized cases of SARS-COV2 infection in Reus, Spain between 15th March 2020 to 30th April 2020. Total of 43 deaths occurred during the study period. Fever, pneumonia, ARDS, diabetes mellitus and cancer were independently associated with mortality.

- While authors present important demographic risk factors, presenting symptoms and comorbidities, in a study where mortality is mainly secondary to respiratory failure data on mechanical ventilation, face mask ventilation/BiPAP, high flow oxygen, number of deaths due to respiratory failure etc should be provided to strengthen the discussion.

- 43 patients died during the study but only 34 were admitted to ICU. Further details for cause of death in these patients should be added. Did all patients die of respiratory failure ? If so, why did all patients who died did not require an ICU admission ? How many were mechanically ventilated ?

- Other factors like anticoagulation, treatments used like corticosteroids etc should also be included.

- Authors should discuss rationale of only choosing charlson’s index and McCabe index but not choosing scoring systems which which predict acuity/severity of acute infection. As SARS-COV2 is an acute infection, charlson’s index and McCabe index do not provide the entire picture and should be integrated with scoring systems which predict severity of acute infection.

- Discussion needs to be strengthened after above are added.

Reviewer #2: This paper presents simple, concise data analyzing the correlation between symptoms and comorbidities and mortality in hospitalized COVID-19 positive patients (n=188) in Reus, Spain between 15th March and 30th April 2020. Of the symptoms and comorbidities recorded in patients, fever, pneumonia, ARDS, Type-2 diabetes, cancer and age independently showed statistically significant association with mortality, adjusted for lifestyle factors.

As the authors of the paper mention, few studies have been published that analyse the risk factors associated with a higher mortality rate due to SARS-CoV-2 infection, outside of China. The study reveals the most vulnerable COVID-19 patients and during this pandemic we face, this knowledge will be crucial to doctors and hospitals treating such at-risk patients. Any study providing clear analytical data on the disease is important to public health and needs to be published given the proclivity of spread of misinformation in the absence of clear expert views.

Study Design- The study design is sound with an appropriate inclusion criterion for patients part of the study. While it may be judged as outside the scope of the study, a possible category of data may have to pertained patients in need of ventilation or oxygen support as opposed to those not requiring respiratory intervention, and their respective mortality rates.

Results- Major results i.e. correlation of age and conditions such as diabetes mellitus and cancer, are in agreement with results of previous studies conducted internationally. Sufficient citation is provided for these results. The authors suggest a possible link between use of corticosteroids and lower incidence of COVID-19 in cancer patients. While the paper they cite refers primarily to the effect methylprednisolone may have on reduced mortality in advanced stages of the disease, a NIHR-funded RECOVERY study, released since, found corticosteroid dexamethasone to similarly reduce mortality in COVID-19 patients requiring ventilation.

Agreeing with the authors, the study has a limited sample size which necessitates evaluating its results alongside results from similar studies to have a truly representative idea of risks influencing mortality in COVID-19 patients.

6. PLOS authors have the option to publish the peer review history of their article (what does this mean?). If published, this will include your full peer review and any attached files.

Reviewer #1: **Yes: **Trushil Shah, MD, MSc

Reviewer #2: **Yes: **Dr. Alok S Shah

---

## [Author Response · Author response to Decision Letter 0]

20 Jul 2020

REVISION NOTE

PONE-D-20-15853

Risk factors associated with mortality in hospitalized patients with SARS-CoV-2 Infection. A prospective, longitudinal, unicenter study in Reus, Spain.

PLOS ONE

EDITORIAL COMMENTS:

Comment #1: “Please ensure that your manuscript meets PLOS ONE's style requirements, including those for file naming. The PLOS ONE style templates can be found at

https://journals.plos.org/plosone/s/file?id=ba62/PLOSOne_formatting_sample_title_authors_affiliations.pdf”

Response: Done.

Comment #2: “Thank you for including your ethics statement: "This study was approved by the Ethics Committee of our Institution (Resolution CEIM 040/2018, amended on 16 April 2020)." a) Please amend your current ethics statement to include the full name of the ethics committee/institutional review board(s) that approved your specific study. b) Once you have amended this/these statement(s) in the Methods section of the manuscript, please add the same text to the “Ethics Statement” field of the submission form (via “Edit Submission”).For additional information about PLOS ONE ethical requirements for human subjects research, please refer to http://journals.plos.org/plosone/s/submission-guidelines#loc-human-subjects-research.”

Response: Done (lines 99 to 101).

Comment #3: “Thank you for stating the following in the Acknowledgments Section of your manuscript:

"This study was supported by a grant from the Fundació la Marató de TV3 (201807-10), Barcelona, Spain." We note that you have provided funding information that is not currently declared in your Funding Statement. However, funding information should not appear in the Acknowledgments section or other areas of your manuscript. We will only publish funding information present in the Funding Statement section of the online submission form. Please remove any funding-related text from the manuscript and let us know how you would like to update your Funding Statement. Currently, your Funding Statement reads as follows: "The funders had no role in study design, data collection and analysis, decision to publish, or preparation of the manuscript."

Response: Done. 

Comment #4: “Please upload a copy of Supporting Information which you refer to in your text on page 12.”

Response: Done (line 112) and Supporting Information file. 

REVIEWER #1:

Comment #1: “Authors present a prospective longitudinal study on 188 hospitalized cases of SARS-COV2 infection in Reus, Spain between 15th March 2020 to 30th April 2020. Total of 43 deaths occurred during the study period. Fever, pneumonia, ARDS, diabetes mellitus and cancer were independently associated with mortality.”

Response: General comment not requiring any specific response.

Comment #2: “While authors present important demographic risk factors, presenting symptoms and comorbidities, in a study where mortality is mainly secondary to respiratory failure data on mechanical ventilation, face mask ventilation/BiPAP, high flow oxygen, number of deaths due to respiratory failure etc should be provided to strengthen the discussion.” 

Response: We accept the Reviewer’s criticism. These data have been added to the manuscript (Table 4 and lines 183 to 189).

Comment #3: “43 patients died during the study but only 34 were admitted to ICU. Further details for cause of death in these patients should be added. Did all patients die of respiratory failure ? If so, why did all patients who died did not require an ICU admission ? How many were mechanically ventilated ?”

Response: Thirty-seven patients died of respiratory failure, 4 of multi-organ failure and 2 of cardiogenic shock. Some seriously ill patients could not be admitted to the ICU due to their pathological history and/or comorbidities associated with their advanced age and who made aggressive treatments inadvisable. These data have been added to the manuscript (lines 121 to 125).

Comment #4: “Other factors like anticoagulation, treatments used like corticosteroids etc should also be included.”

Response: We accept the Reviewer’s criticism. These data have been added to the manuscript (Table 4 and lines 189 to 191).

Comment #5: “Authors should discuss rationale of only choosing charlson’s index and McCabe index but not choosing scoring systems which which predict acuity/severity of acute infection. As SARS-COV2 is an acute infection, charlson’s index and McCabe index do not provide the entire picture and should be integrated with scoring systems which predict severity of acute infection.”

Response: We chose the McCabe and Charlson indices because they are easy to apply in all patients, both in the less severe, and in those receiving palliative treatment or admitted to the Social Health Unit. Other scores are difficult to apply in less severe patients. For example, the CURB 1 scale only applies to patients with pneumonia. The quick SOFA index requires measurement of respiratory rate, a data not collected in milder patients, and SOFA and SAPS-II require arterial blood gas analysis, which is difficult to justify in patients not admitted to the ICU. This aspect has been commented in the Discussion (lines 258 to 263 and new refs. #41-43).

Comment #6: “Discussion needs to be strengthened after above are added.”

Response: We have modified the Discussion accordingly (lines 207 to 210, 235 to 237, amd 258 to 263).

REVIEWER #2:

Comment #1: “This paper presents simple, concise data analyzing the correlation between symptoms and comorbidities and mortality in hospitalized COVID-19 positive patients (n=188) in Reus, Spain between 15th March and 30th April 2020. Of the symptoms and comorbidities recorded in patients, fever, pneumonia, ARDS, Type-2 diabetes, cancer and age independently showed statistically significant association with mortality, adjusted for lifestyle factors.”

Response: General comment not requiring any specific response.

Comment #2: “As the authors of the paper mention, few studies have been published that analyse the risk factors associated with a higher mortality rate due to SARS-CoV-2 infection, outside of China. The study reveals the most vulnerable COVID-19 patients and during this pandemic we face, this knowledge will be crucial to doctors and hospitals treating such at-risk patients. Any study providing clear analytical data on the disease is important to public health and needs to be published given the proclivity of spread of misinformation in the absence of clear expert views.”

Response: We thank the Reviewer for his kind words about our work.

Comment #3: “Study Design- The study design is sound with an appropriate inclusion criterion for patients part of the study. While it may be judged as outside the scope of the study, a possible category of data may have to pertained patients in need of ventilation or oxygen support as opposed to those not requiring respiratory intervention, and their respective mortality rates.”

Response: We agree with the Reviewer. These data have been added to the revised manuscript (Table 4 and lines 183 to 189).

Comment #4: “Results- Major results i.e. correlation of age and conditions such as diabetes mellitus and cancer, are in agreement with results of previous studies conducted internationally. Sufficient citation is provided for these results. The authors suggest a possible link between use of corticosteroids and lower incidence of COVID-19 in cancer patients. While the paper they cite refers primarily to the effect methylprednisolone may have on reduced mortality in advanced stages of the disease, a NIHR-funded RECOVERY study, released since, found corticosteroid dexamethasone to similarly reduce mortality in COVID-19 patients requiring ventilation.”

Response: We thank the Reviewer for this suggestion. We have added a reference on the RECOVERY study (Ref. #33). However, as a consequence of a Reviewer # 1 criticism, we have analyzed the influence of corticosteroid treatment on our population and have found no significant differences. Because of this we have modified our comment (lines 235 to 237).

Comment #5: “Agreeing with the authors, the study has a limited sample size which necessitates evaluating its results alongside results from similar studies to have a truly representative idea of risks influencing mortality in COVID-19 patients.”

Response: We fully agree with this comment.

---

## [Decision Letter · Decision Letter 1]

24 Aug 2020

Risk factors associated with mortality in hospitalized patients with SARS-CoV-2 infection. A prospective, longitudinal, unicenter study in Reus, Spain

PONE-D-20-15853R1

Dear Dr. Camps,

We’re pleased to inform you that your manuscript has been judged scientifically suitable for publication and will be formally accepted for publication once it meets all outstanding technical requirements.

Kind regards,

Muhammad Adrish

Academic Editor

PLOS ONE

Additional Editor Comments (optional):

Reviewers' comments:

Reviewer's Responses to Questions

**Comments to the Author**

1. If the authors have adequately addressed your comments raised in a previous round of review and you feel that this manuscript is now acceptable for publication, you may indicate that here to bypass the “Comments to the Author” section, enter your conflict of interest statement in the “Confidential to Editor” section, and submit your "Accept" recommendation.

Reviewer #1: All comments have been addressed

2. Is the manuscript technically sound, and do the data support the conclusions?

Reviewer #1: Yes

3. Has the statistical analysis been performed appropriately and rigorously? 

Reviewer #1: Yes

4. Have the authors made all data underlying the findings in their manuscript fully available?

Reviewer #1: Yes

5. Is the manuscript presented in an intelligible fashion and written in standard English?

Reviewer #1: Yes

6. Review Comments to the Author

Reviewer #1: (No Response)

7. PLOS authors have the option to publish the peer review history of their article (what does this mean?). If published, this will include your full peer review and any attached files.

Reviewer #1: **Yes: **Trushil G. Shah, MD, M.Sc

---

## [Editor Report · Acceptance letter]

26 Aug 2020

PONE-D-20-15853R1 

Risk factors associated with mortality in hospitalized patients with SARS-CoV-2 infection. A prospective, longitudinal, unicenter study in Reus, Spain 

Dear Dr. Camps:

I'm pleased to inform you that your manuscript has been deemed suitable for publication in PLOS ONE. Congratulations! Your manuscript is now with our production department. 

Kind regards, 

on behalf of

Dr. Muhammad Adrish 

Academic Editor

PLOS ONE